# Strategic Planning of a Joint SARS-CoV-2 and Influenza Vaccination Campaign in the UK

**DOI:** 10.3390/vaccines12020158

**Published:** 2024-02-03

**Authors:** Dauda Ibrahim, Zoltán Kis, Maria M. Papathanasiou, Cleo Kontoravdi, Benoît Chachuat, Nilay Shah

**Affiliations:** 1The Sargent Centre for Process Systems Engineering, Department of Chemical Engineering, Imperial College London, London SW7 2AZ, UK; z.kis10@imperial.ac.uk (Z.K.); maria.papathanasiou11@imperial.ac.uk (M.M.P.); cleo.kontoravdi98@imperial.ac.uk (C.K.); b.chachuat@imperial.ac.uk (B.C.); n.shah@imperial.ac.uk (N.S.); 2Department of Chemical and Biological Engineering, The University of Sheffield, Sheffield S1 3JD, UK

**Keywords:** SARS-CoV-2 vaccine, influenza vaccine, vaccination campaign, economic analysis, mathematical programming

## Abstract

The simultaneous administration of SARS-CoV-2 and influenza vaccines is being carried out for the first time in the UK and around the globe in order to mitigate the health, economic, and societal impacts of these respiratory tract diseases. However, a systematic approach for planning the vaccine distribution and administration aspects of the vaccination campaigns would be beneficial. This work develops a novel multi-product mixed-integer linear programming (MILP) vaccine supply chain model that can be used to plan and optimise the simultaneous distribution and administration of SARS-CoV-2 and influenza vaccines. The outcomes from this study reveal that the total budget required to successfully accomplish the SARS-CoV-2 and influenza vaccination campaigns is equivalent to USD 7.29 billion, of which the procurement costs of SARS-CoV-2 and influenza vaccines correspond to USD 2.1 billion and USD 0.83 billion, respectively. The logistics cost is equivalent to USD 3.45 billion, and the costs of vaccinating individuals, quality control checks, and vaccine shipper and dry ice correspond to USD 1.66, 0.066, and 0.014, respectively. The analysis of the results shows that the choice of rolling out the SARS-CoV-2 vaccine during the vaccination campaign can have a significant impact not only on the total vaccination cost but also on vaccine wastage rate.

## 1. Introduction

The World Health Organization (WHO) declared Severe Acute Respiratory Syndrome Corona Virus 2 (aka corona virus disease 2019 or COVID-19) a pandemic on 11 March 2020 [1,2,3,4] after the emergence of the virus in Wuhan, China. COVID-19 has claimed the lives of around 6.2 million individuals globally, and around 58.1 million individuals are still infected with the highly contagious virus [5,6,7]. Even though governments and non-governmental organisations have made funds available to fight COVID-19 [8,9,10,11,12], and the global vaccination programmes against COVID-19 have been largely successful, especially in high-income countries [13,14,15,16,17,18], the disease is still prevalent around the globe [19,20,21]. This is partly due to the emergence of new variants (e.g., Alpha (B.1.1.7), Beta (B.1.351), Delta (B.1.617.2), and Omicron (B.1.1.529)) [22,23,24,25] that render the current vaccines less effective. Booster jabs (e.g., BNT162b2 developed by Pfizer and BioNTech [26,27,28] and mRNA-1273 developed by Moderna [29,30,31]) have been developed and administered to high-risk and vulnerable individuals to avoid developing complications that could lead to hospitalisation or even death. Similarly, the influenza virus (e.g., A(H1N1), A(H3N2), and B/Victoria lineage) [32,33,34,35] has been around for some time, and high-risk and vulnerable individuals are required to take an annual flu jab to boost immunity and to reduce the impact of the disease within society. For the first time, SARS-CoV-2 and influenza vaccines are distributed and administered simultaneously, posing significant logistical and distribution challenges. For example, SARS-CoV-2 and influenza vaccines require different types of storage technologies, especially RNA-based SARS-CoV-2 vaccines that need to be stored and transported at ultra-low temperature (ULT) conditions. Consequently, an effective and efficient planning method would be required for the vaccination campaigns to be successful.

During a vaccination campaign, vaccines are delivered through complex supply chains, which typically comprise manufacturing facilities, intermediate storage locations such as warehouses and vaccine stores, and administration points (GP surgeries, hospitals, pharmacies, clinics, and mass vaccination centres). The design and planning of vaccine supply chains require optimal selection of storage locations, production planning at manufacturing plants, inventory management, distribution planning, storage capacity planning, selection of routes and transport types, etc. In previous work, a general purpose simulation-based analytical tool known as HERMES—Highly Extensible Resource for Modeling Supply Chains was used to (i) assess the performance of a vaccine supply chain [36,37,38,39], (ii) plan the introduction of new vaccines against rotavirus and pneumococcus [40], and (iii) redesign a vaccine supply chain to allow access of vaccines during routine and supplementary vaccination campaigns in low- and middle-income countries [41,42,43]. However, HERMES does not support the optimisation of supply chains, leading to solutions that could be suboptimal. Cavalho et al. [44] considered three performance indicators (economic, environmental, and social performance) to develop a multi-objective mixed-integer linear programming (MO-MILP) model for the optimal design and planning of a sustainable vaccine supply chain. A case study on the distribution of vaccines against poliomyelitis and MMR (measles, mumps, and rubella) within Europe indicated that the proposed model is capable of obtaining solutions that are economically relevant and would also lead to a low global warming potential. Kis et al. [45] proposed a steady-state MILP model for the distribution of vaccine candidates developed using various platform technologies such as RNA vaccines, outer membrane vesicle vaccines with genetically customisable membrane antigens, virus-like particle vaccines with genetically configurable epitopes, and humanised yeast-produced vaccines. The model indicated both the supply chain configuration and delivery type that would lead to a maximum net present value.

More recently, Georgiadis and Georgiadis [46] addressed the distribution of SARS-CoV-2 vaccines in Greece. The MILP model proposed by the authors takes into account the storage, distribution, and administration aspect of the vaccine supply chain. However, their model does not account for quality control (QC) checks; fill-finish plants; production planning; selection of transport mode; and, most importantly, management of vaccine thermal shippers. Ibrahim et al. [47] proposed a novel supply chain model that addresses the complexities related to the supply and distribution of vaccine candidates developed using the most advance platform technologies. By considering the essential components of the supply chain (such as manufacturing and fill-finish plants, storage locations, administration points, transport modes, quality control checks, and management of thermal shippers), the authors developed an MILP supply chain model for the distribution and administration of RNA SARS-CoV-2 vaccines (BNT162b2—Comirnaty^®^) in the UK. The outcomes from this work indicated that the proposed model can identify economically optimal supply chains in addition to revealing locations where stockouts could occur should there be a shortage in vaccine supply. The authors extended the model to incorporate SARS-CoV-2 vaccines developed using other platform technologies, such as viral vectors (AZD1222—Vaxzevria^®^), in addition to RNA vaccines (BNT162b2—Comirnaty^®^) [48]. None of the aforementioned works have addressed the challenges involved in planning of simultaneous SARS-CoV-2 and influenza vaccination campaigns.

This work develops a novel multi-product MILP vaccine supply chain model that can be used to plan and optimise the simultaneous distribution and administration of SARS-CoV-2 and influenza vaccines. In addition to planning vaccine administration, the supply chain model (i) facilitates the transport of vaccines from manufacturing plants to administration points; (ii) sets out an effective vaccination strategy focusing on the most vulnerable segment of a country-wide population; (iii) assesses workforce requirements and financial planning; and (iv) monitors the progress of a vaccination and identifies resources needed throughout the vaccination period, including storage equipment (fridges, freezers, and ultra-low-temperature freezers), transport modes (planes, trucks, and vans), vaccine doses (SARS-CoV-2 and influenza vaccines), thermal shippers and dry ice, and healthcare personnel. The multi-product MILP model minimises the total cost incurred over the entire vaccination period while setting targets for total vaccine doses needed; determining storage capacity requirements at the central store, regional stores, and administration points; and identifying cost-effective transportation routes. The capability of the proposed model is demonstrated using a real-world case study concerned with SARS-CoV-2 and influenza vaccination campaigns in the UK.

The remainder of this article is organised as follows. Section 2 presents the proposed vaccine supply chain model. Section 3 presents the case study information, while Section 4 presents and discusses the outcomes from this work. Lastly, Section 5 concludes the paper and presents future work.

## 2. Method: Vaccine Supply Chain Modelling

### 2.1. Structure of SARS-CoV-2 and Influenza Vaccines’ Supply Chains

The new multi-product MILP vaccine supply chain model was developed using the concept of a multi-echelon supply chain, in which each echelon denotes a supply chain level. The structure of the vaccine supply chain considered in this work consists of four levels: manufacturing plants or imports, central warehouse, regional warehouse, and administration points (GP surgeries, hospitals, pharmacies, clinics, and mass vaccination centres), as shown in Figure 1. The manufacturing plants produce various types of SARS-CoV-2 and influenza vaccine candidates, either in-country or overseas, whilst central and regional warehouses store vaccines temporarily using suitable cooling technologies such as fridges, freezers, and ultra-low-temperature freezers. Lastly, vaccines are injected into people’s arms at administration points. The domestic transportation of vaccines between supply chain entities is carried out using either a refrigerated van or a refrigerated truck, while imports are carried out using an airplane. The mathematical formulation representing the proposed vaccine supply chain structure is presented in Section 2.1 and Section 2.2.

### 2.2. SARS-CoV-2 and Influenza Vaccine Supply Chain Model

#### 2.2.1. Objective Function

The mathematical formulation representing the SARS-CoV-2 and influenza vaccine supply chain consists of an objective function and constraints. The objective function denotes the key performance indicator to be either minimised or maximised, for example, minimisation of cost and vaccine wastage or maximisation of net present value, profit, vaccination coverage, etc. The equations and detailed description of the mathematical formulation can be found in Appendix A. In this work, the objective function considered is the total cost incurred over the entire vaccination period, which is the sum of the annualised installed capital cost of the supply chain equipment and building, total cost of operating the equipment and building throughout the vaccination period, transportation cost, and miscellaneous cost.

The miscellaneous cost includes the cost of administering vaccines to target patients, vaccine procurement, quality control checks, thermal shippers, and dry ice. On the other hand, the annualised installed capital cost is the sum of the installed cost of supply chain assets, including storage devices (fridges, freezers, and ultra-low-temperature freezers) at central and regional warehouses and building infrastructure, while the operating cost is defined as the vaccine inventory at a given time period multiplied by the unit inventory holding cost. Other cost components calculated by the model include:Total transportation cost between supply chain entities, which is defined as the unit transportation cost for a specific transport mode multiplied by the number of trips covered at a given time period. Note that the unit transportation cost accounts for travel distance, driver wages, the fuel and vehicle maintenance cost, and the annualised capital cost of the transport vehicle.Total cost of dry ice is calculated as the total dry ice required multiplied by the unit price of dry ice. Note that each full shipper leaving warehouses is loaded with sufficient dry ice to maintain the sub-zero storage temperature of −80 °C.The cost of vaccine shippers is estimated by multiplying the number of shippers supplied at the start of the vaccination campaign by the unit price of each shipper. Here, it is assumed that no shipper will be damaged and/or replaced throughout the vaccination campaign.

#### 2.2.2. Model Constraints

The flow of vaccines to and from each entity of the vaccine supply chain is defined by the material balance constraints. As shown in Figure 1, the entities of the supply chain include manufacturing plants, warehouses, regional stores, and administration points. Equations are used to define the inventory of vaccines for these entities. In warehouses, vaccines are expected to undergo quality control (QC) checks, which can take up to two weeks. To account for the delay due to QC, the inventory of vaccines at warehouses is partitioned into two segments, i.e., vaccine inventory before and after QC. The inventory of vaccines before QC is calculated by subtracting the quantity of vaccines leaving for QC checks from the sum of vaccines arriving from fill-finish plants, oversees imports, and the vaccine inventory in the previous time period. The inventory of vaccines after QC is calculated by subtracting the quantity of vaccines leaving warehouses from vaccines that have undergone QC checks and vaccine inventory at the previous time period. Here, each time period is equivalent to one week.

For the sake of brevity, the equations denoting the material balances over manufacturing plants, warehouses, regional stores, and administration points are presented in Appendix A. In addition, equations denoting the following constraints are also presented in the Appendix A:Tracking vaccine shippers: warehouses–regional store–clinics–warehouses;Shelf life of vaccines at administration points;Safety stock, maximum inventory, and bounds on QC checks, production rate, and import rate;SARS-CoV-2 and influenza vaccines demand;Number of trips.

In this work, the proposed multi-product MILP vaccine supply chain model is implemented in the GAMS software tool version 30.3.0 [49], and a case study on simultaneous SARS-CoV-2 and influenza vaccination campaigns is used to demonstrate the capabilities of the proposed model, as shown in Section 3 and Section 4.

## 3. Application: SARS-CoV-2 and Influenza Vaccination Campaign in the UK

The capabilities of the proposed multi-product MILP vaccine supply chain model are demonstrated using real-world data on a joint SARS-CoV-2 and influenza vaccination campaign in the UK. The case study data were collected from several reliable sources, and they include information on vaccine distribution network in the UK, target population, and SARS-CoV-2 and influenza vaccine candidates.

### 3.1. Target Population and Vaccine Candidates

Appendix A show the target population recommended by the Joint Committee on Vaccination and Immunisation (JCVI) for SARS-CoV-2 and influenza vaccination. Both cohorts were selected according to risk factors and vulnerability against the diseases. Similarly, Appendix A presents the vaccine candidates recommended for SARS-CoV-2 and influenza vaccination campaigns.

### 3.2. Vaccine Distribution Network in the UK

The structure of the vaccine supply chain considered here is similar to the one presented in Ref [47] but includes multiple manufacturing plants for SARS-CoV-2 and influenza vaccines. Influenza vaccines Flucelvax Tetra^®^ and Fluad^®^ are manufactured by Seqirus, while Fluenz Tetra^®^ is manufactured by AstraZeneca; both plants are located in Liverpool, UK. Vaxigrip Tetra^®^ and Influvac Tetra^®^ are manufactured by Sanofi Pasteur and Mylan located in Val-de-Reuil, France, and Olst, the Netherlands, respectively. Similarly, SARS-CoV-2 vaccines Comirnaty^®^ (BNT162b2) and Spikevax^®^ (mRNA-1273) are manufactured by Pfizer-BioNTech and Lonza located in Puur, Belgium, and Basel, Switzerland, respectively. Vaccines manufactured overseas are airlifted to four central warehouses in the UK located in London, Edinburgh, Cardiff, and Belfast. Subsequently, the vaccines are distributed to 12 regional warehouses, of which nine are located in England (North East, North West, Yorkshire and the Humber, East Midlands, West Midlands, East of England, London, South East, South West) and one each in Edinburgh, Cardiff, and Belfast. Within the UK, vaccines are administered at hospitals, clinics, GP surgeries, vaccination centres, and care homes. The administration points are spatially distributed across the UK following the policy enacted by the UK government to establish vaccination centres within a 10-mile radius of a populace [50]. In this work, the administration points in each region are grouped into a cluster, leading to 12 clusters—i.e., one in each region. This approach can reduce the computational burden encountered during simulation and optimisation of the supply chain network.

The central warehouses, regional stores, and administration points consist of three types of storage technologies, i.e., a fridge, freezer, and ultra-low-temperature freezer. The actual total capacity of storage devices required is determine by the planning model; however, the storage technology at each location has a maximum capacity. Appendix A shows the maximum capacities of storage technologies at central and regional warehouses as well as administration points. Using the maximum capacities, we estimated the installation and operating cost of each storage technology, as shown in Appendix A. The installation cost takes into account the purchase cost of cooling devices and building cost, while the operating cost accounts for utility bills and staff wages. On arrival at central warehouses, vaccines undergo quality control (QC) checks, which can take up to two weeks. The cost of QC checks is estimated following information provided by the National Institute for Biological Standards and Control (NIBSC) [51].

The domestic distribution of vaccines is carried out using a refrigerated truck and a refrigerated van, while imports are carried out using an airplane. The unit transport cost and capacity of each transport mode is shown in Appendix A. The unit transport cost takes into account driver wages, fuel cost, the annualised capital cost of each vehicle, and average annual distance covered. The travel distance is estimated using Google maps, assuming straight line distances between supply chain entities.

## 4. Results

### 4.1. SARS-CoV-2 and Influenza Vaccines Demand Planning

Across the UK, a total of ≈55 million and ≈48 million individuals are to be vaccinated against SARS-CoV-2 and the influenza virus, respectively, as shown in Appendix A. The SARS-CoV-2 vaccination campaign focuses on vaccinating care home residents; residential care workers; individuals 80+ years of age; healthcare workers; social care workers; individuals 75–79 years of age; individuals 70–74 years of age; clinically extremely vulnerable individuals (under 70); individuals 65–69 years of age; at risk individuals (under 65); and individuals 60–64, 55–59, 50–54, 18–49, and 16–17 years of age, in that order [52,53]. Similarly, the influenza vaccination campaign aims to vaccinate individuals 65+ years of age; at risk individuals (under 65); pregnant women; individuals 50– 64 years of age; care home residents; residential care workers; healthcare workers; social care workers; individuals ages 2 and 3; and school-age children: Reception (age 4–5), Year 1 (age 5–6), Year 2 (age 6–7), Year 3 (age 7–8), Year 4 (age 8–9), Year 5 (age 9–10), Year 6 (age 10–11), and Year 7 (age 11–12), in that order [54,55,56].

Apart from age groups 16 to 49 (arriving for SARS-CoV-2 vaccine only) and age groups 2 to 12 (arriving for influenza vaccines only), the individuals arriving at vaccination centres for the two campaigns are the same. By dividing the population of each cohort by vaccination rate across the UK, it is possible to estimate the time period required to vaccinate each cohort as well as the time frame to complete the vaccination exercise, which is around 28 weeks for this study. Therefore, the time frame to complete the vaccination exercise is 30 weeks, considering that each batch of SARS-CoV-2 and influenza vaccines will undergo quality control checks at central warehouses, which takes up to two weeks. Only batches that meet the quality control procedures (as set out by the Medicines and Healthcare Regulatory Agency—MHRA) are used during the vaccination campaign. In this study, it was assumed that all batches of both SARS-CoV-2 and influenza vaccines satisfy the quality control checks; hence, no vaccines are discarded at this stage. Figure 2 shows the doses of SARS-CoV-2 and influenza vaccines required over the complete vaccination period.

In Figure 2, the demands for SARS-CoV-2 and influenza vaccines vary throughout the vaccination period. This results from the fact that the cohort for the two vaccination campaigns differs slightly. There is no supply of SARS-CoV-2 vaccines in Weeks 1 and 2, as the vaccination starts two weeks after the commencement of influenza vaccination. Furthermore, SARS-CoV-2 vaccination ends three weeks earlier than influenza. No influenza vaccine is supplied during Weeks 20 to 23 since no influenza vaccination is conducted during this period according to the scheduling of patients carried out in this study.

### 4.2. Optimal Planning of Joint SARS-CoV-2 and Influenza Vaccination

The demand profile shown in Figure 2 together with information on recommended vaccine candidates and target individuals in Appendix A are applied on the supply chain model proposed in Section 2 in order to plan and optimise costs associated with the joint SARS-CoV-2 and influenza vaccination campaigns as well as to establish a cost-effective investment strategy. The results from the planning and optimisation study are shown in Table 1 and Figure 3.

The total cost of vaccinating all target individuals against SARS-CoV-2 and influenza virus is USD ≈ 7.2 billion, which includes the cost of vaccine shippers and dry ice, vaccine administration, vaccine procurement, quality control, and logistics. As can be seen, the logistics cost is the largest cost component, followed by the cost of vaccine procurement, whilst the cost of shippers and dry ice are the smallest. The breakdown of the vaccine procurement cost in Figure 3 shows that the cost of procuring SARS-CoV-2 vaccines is greater than all influenza vaccines combined, which is caused by the high cost per dose of SARS-CoV-2 vaccines compared to influenza vaccines.

The quantity of BNT162b2 SARS-CoV-2 vaccines required to fulfil demand is estimated at ≈54.5 million doses, corresponding to USD 2.1 billion. Similarly, the demand for Fluad^®^, Flucelvax Tetra^®^, Vaxigrip Tetra^®^, Fluenz Tetra^®^, and Influvac Tetra^®^ corresponds to 12.03, 19.99, 0.015, 6.31, and 1.62 million doses and procurement costs of USD 0.269, 0.302, 0.0005, 0.212, and 0.048 billion, respectively. Figure 4a,b shows the breakdown of the total logistics cost and transportation cost, respectively.

In addition to the transportation cost, the logistics cost comprises the capital and operating cost. Figure 5a,b shows the cost required to install and operate the various types cooling devices (fridges, freezers, and ultra-low-temperature freezers) at central warehouses, regional stores, and administration points. As can be seen, the cost of installing and operating fridges constitutes the largest portion of capital and operating cost, followed by ULT freezers and lastly conventional freezers. This is because five out of the seven vaccines used during the vaccination campaign require cooling at 2–8 °C, leading to a total cooling space requirement of 958,173 L (corresponding to 3194 fridges of size 300 L). Only the BNT162b2 SARS-CoV-2 vaccine requires ULT cooling, leading to a total cooling space requirement of 86,430 L (corresponding to 288 ULT freezers of size 300 L). Lastly, the mRNA1273 SARS-CoV-2 vaccine requires cooling at −24 °C, leading to a cooling space requirement of 20,203 L (corresponding to 67 freezers of size 300 L).

### 4.3. Impact of SARS-CoV-2 Vaccine Type on Logistics and Total Cost

During the planning and optimisation of costs associated with the joint SARS-CoV-2 and influenza vaccination campaign, the supply chain model is allowed to select the SARS-CoV-2 vaccine type to be supplied and administered to target individuals in line with the current UK vaccination policy, in which either the BNT162b2 or mRNA-1273 SARS-CoV-2 vaccine is administered to patients [52,53]. In Section 4.2, the supply chain model selected BNT162b2 over mRNA-1273 as the optimal SARS-CoV-2 vaccine to be used during the vaccination campaign. It should be noted that the optimal SARS-CoV-2 vaccine selected by the optimisation model is purely based on cost, without considering clinical information at this stage. Here, the impact of SARS-CoV-2 vaccine type on the total vaccination cost is investigated. Figure 6 shows the optimal results when the UK government decides to cover 20%, 40%, 60%, 80%, and 100% of the SARS-CoV-2 vaccination with mRNA-1273.

As can be seen, the total vaccination cost increases with the increasing percentage of mRNA-1273 used during the vaccination campaign. The rapid increase in cost is due to the high procurement cost of mRNA-1273 (USD 34.84 dose^−1^) compared to BNT162b2 (USD 18.66 dose^−1^). This point can be explained by visualising the cost breakdown of Scenarios A and B, as shown in Figure 7.

### 4.4. Vaccine Administration at Clinics, Hospitals, GP Surgeries, and Vaccination Centres

The simultaneous administration of SARS-CoV-2 and influenza vaccines is being carried out for the first time in the UK; consequently, proper planning is needed across all administration points to ensure that target patients are given the recommended vaccine dose regimen with the minimum overall vaccine wastage. Figure 8 shows the progress of SARS-CoV-2 and influenza vaccination in Cardiff; Edinburgh; Belfast; and the nine regions of England: North East, North West, Yorkshire and the Humber, East Midlands, West Midlands, East of England, London, South East, and South West.

In Figure 8, the influenza vaccination starts with care home residence and residential care workers followed by individuals ages 65 plus, healthcare workers, social care workers, at risk individuals (under 65), individuals ages 50–64, pregnant women, indivudlas ages 2 and 3, and individuals ages 4 to 11. On the other hand, SARS-CoV-2 vaccination covers only care home residence, residential care workers, individuals ages 65 plus, healthcare workers, social care workers, at risk individuals, individuals ages 50–64, pregnant women, individuals ages 18 to 49, and individuals ages 16 to 17, as per the UK government recommendation [54,55,56]. Note that the scheduling of appointments for individuals arriving at vaccine administration points follows the aforementioned order.

## 5. Discussion

To successfully accomplish the SARS-CoV-2 and influenza vaccination campaigns, sufficient doses of the recommended vaccines must be made available in order to satisfy demand, i.e., the number of patients arriving at vaccination centres to be inoculated with the vaccines. The vaccine demand profile shown in Figure 2 is an important tool that can be utilised by the government and policy makers to set out vaccine procurement strategies ahead of the simultaneous SARS-CoV-2 and influenza vaccination campaigns. By December 2021, the UK government secured the supply of 114 million doses of SARS-CoV-2 vaccines to cover SARS-CoV-2 vaccination campaigns in Years 2022 and 2023 [57,58,59]. The procured vaccine doses consist of 54 million BNT162b2 and 60 million mRNA-1273. As can be deduced from Figure 2, the total doses required for the SARS-CoV-2 vaccination campaign is ≈55 million doses, which is 52% lower than the total doses procured. Hence, the total doses procured by the UK government are sufficient to cover two SARS-CoV-2 vaccination campaigns as expected. Looking forward, the analysis presented here can be used to update the current SARS-CoV-2 vaccine demand, especially when the UK government and policy makers decide to add more cohorts to the existing ones.

From the economic analysis in Section 4.2, the overall cost of vaccinating target individuals against SARS-CoV-2 and the influenza virus is approximately 7.2 billion USD, of which the logistics cost constitutes the largest percent (47%), followed by the vaccine procurement cost (29%). The logistics cost is the sum of the total capital cost, total operating cost, and total transportation cost. As can be seen in Figure 4a,b, the cost to be invested in cold chain infrastructure dominates the logistics cost of the combined SARS-CoV-2 and influenza vaccines supply chain, while the transportation investment is the smallest component. Furthermore, Figure 4b indicates that the cost of transporting SARS-CoV-2 vaccines contributes around 77.1% of the total transportation cost. The SARS-CoV-2 vaccines BNT162b2 and mRNA-1273 are manufactured by Pfizer-BioNTech and Lonza, with the manufacturing plants located in Puur, Belgium, and Basel, Switzerland, respectively. These vaccine candidates are airlifted to central warehouses in the UK (London, Edinburg, Cardiff, and Belfast) before distribution to regional stores and later to administration points, consequently leading to the high cost of transportation compared with influenza vaccine candidates manufactured in the UK, which do not need air transport.

Investigation of the impact of SARS-CoV-2 vaccine type on total vaccination cost indicates that the vaccination cost of BNT162b2 is 14% lower than mRNA-1273, even though BNT162b2 requires a more expensive cooling technology (ULT freezer operating at −80 °C) as well as thermal shippers and dry ice. This analysis shows that the procurement cost of a more thermostable RNA SARS-CoV-2 vaccine outweighs the cost savings resulting from the use of moderate cooling technology (conventional freezer operating at −24 °C) during the storage and transport of SARS-CoV-2 vaccines. Although BNT162b2 has the lowest total cost, its supply chain is more complex and difficult to operate, requiring a system for the effective management of thermal shippers and dry ice as well as temperature monitoring during storage and transportation in order to control temperature excursions that could compromise the potency of the vaccines. Owing to the challenges in handling BNT162b2, this vaccine type is likely to lead to more vaccine wastage compared to mRNA-1273, as reports have indicated several million doses of SARS-CoV-2 vaccines have already been wasted [60,61,62,63]. Once the vial of BNT162b2 is thawed at room temperature, it is expected to be used within two hours; afterwards, any unused doses must be discarded as waste. mRNA-1273 can last for up to twelve hours at room temperature. Also, vials of BNT162b2 can be stored for only five days at 2–8 °C compared to 30 days for mRNA-1273. From the logistics viewpoint, the longer shelf life at 2–8 °C means large quantities of vaccines can be stored at vaccine administration points, thereby decreasing delivery frequency and consequently the logistics cost. This point is reflected in Figure 7, which shows that the logistics cost of BNT162b2 and mRNA-1273 constitute 29.7% and 25.7% of the total cost, respectively.

Ahead of the simultaneous SARS-CoV-2 and influenza vaccination campaign, the curve shown in Figure 6 can be used to estimate the total vaccination cost depending on the type of SARS-CoV-2 vaccine the UK government intends to roll out. This information is vital and can facilitate efficient and effective logistical and financial planning, ensuring that the contract for the supply of vaccines is within the overall budget.

There is no vaccination administration in Weeks 1 and 2, as the initial batches of the vaccines undergo quality control checks at central warehouses. Recall that it was assumed that SARS-CoV-2 vaccination commences two weeks after influenza, meaning individuals inoculated with flu jabs in Week 3 will return to the administration point for their SARS-CoV-2 vaccine in Week 5. Thus, in Figure 8, both SARS-CoV-2 and influenza vaccines are administered between Weeks 5 to 27. Apart from showing the progress of the vaccination campaign, Figure 8 can be used to estimate the approximate capacity of vaccine administration points in the twelve regions of the UK. The capacity required to ensure a successful vaccination campaign in Cardiff, Edinburgh, Belfast, North East, North West, Yorkshire and the Humber, East Midlands, West Midlands, East of England, London, South East, and South West correspond to 0.212, 0.573, 0.418, 0.376, 0.447, 0.479, 0.698, 0.714, 0.475, 0.315, 0.673, and 0.256 million doses per week, respectively. Prior to the joint SARS-CoV-2 and influenza vaccination campaigns, this information can be used to plan the capacities of administration points in order to accommodate the number of individuals arriving per week. Failure to provide enough capacity could lead to a drop in the vaccination rate and immunisation coverage, which could lead to a surge in infection rates, hospitalisations, and deaths.

Another important planning aspect of a vaccination campaign is the human resources required to administer vaccines. For the simultaneous SARS-CoV-2 and influenza vaccination, the workforce requirement is estimated by assuming that administration staff work for six hours per day and take about ten minutes to vaccinate each patient. As expected, southeast England (range: 3488–19,826) requires the largest number of staff to keep up with the large number of patients arriving at vaccination centres each week. The weekly staff requirement of the remaining administration points are as follows: North East (range: 965–5899), North West (range: 2498–15,909), Yorkshire and the Humber (range: 1900–11,625), East Midlands (range: 1690–10,453), West Midlands (range: 2077–12,424), East of England (range: 2382–13,302), London (range: 2068–19,380), South West (range: 2377–13,192), Wales (range: 1118–8760), Scotland (range: 2083–18,704), and Northern Ireland (range: 1248–7099). This information can be used by the government and policy makers to assess the existing workforce in each region prior to the vaccination exercise. Regions with staff deficits can either embark on recruiting additional staff or seek redeployment of staff from regions with a surplus number of staff.

## 6. Conclusions

This work develops a novel multi-product mixed-integer linear programming (MILP) vaccine supply chain model that can be used to plan and optimise the simultaneous distribution and administration of SARS-CoV-2 and influenza vaccines. The model considers the essential features of a typical vaccine supply chain, such as manufacturing and fill-finish plants, storage locations, administration points, transport modes, quality control checks, and management of thermal shippers. Overall, this work has addressed the following challenges related to joint SARS-CoV-2 and influenza vaccination: optimal selection of storage locations, production planning at manufacturing plants, inventory management, distribution planning, storage capacity planning, selection of routes and transport types, etc. The optimisation studies indicate that by minimising total vaccination cost, it is possible to identify cost-effective candidate vaccine supply chains and their associated resource requirements, such as cold chain equipment (fridges, freezers, and ultra-low-temperature freezers), transport devices, healthcare personnel, and doses of recommended vaccine candidates. For the joint SARS-CoV-2 and influenza vaccination campaigns in the UK, the logistics cost dominates the total vaccination cost, followed by the vaccine procurement cost. In addition, the type of SARS-CoV-2 vaccine candidate the UK government decides to deploy for the vaccination campaign can have a significant impact of total vaccination cost, with BNT162b2 leading to a lower total cost compared to mRNA-1273.

## Figures and Tables

**Figure 1 vaccines-12-00158-f001:**
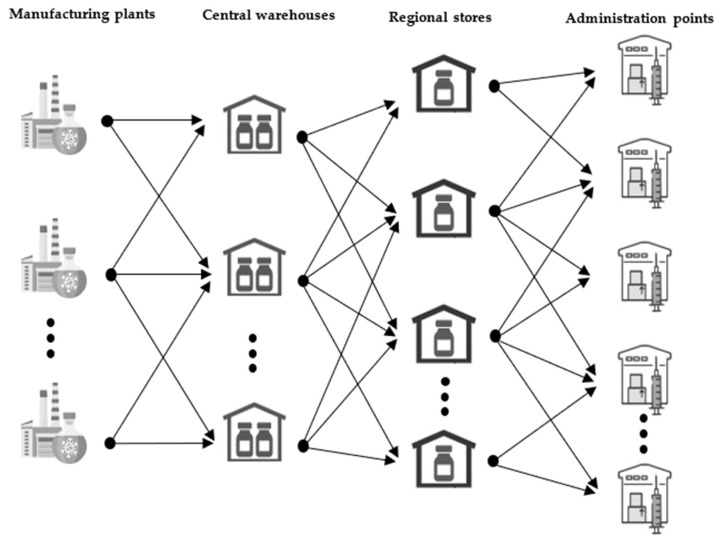
Structure of the supply chain for SARS-CoV-2 and influenza vaccine candidates. The distribution chain consists of manufacturing plants, central and regional warehouses, and administration points (hospitals, GP surgeries, care homes, pharmacies, and vaccination centres). The manufacturing plants can be located in the country or overseas, and vaccines flow from the plants down to the administration points without reverse flow.

**Figure 2 vaccines-12-00158-f002:**
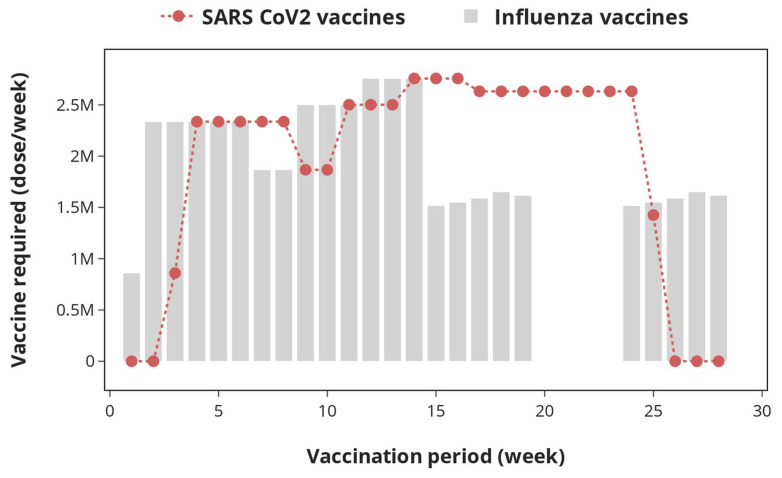
Demand profile for SARS-CoV-2 and influenza vaccines over the entire vaccination period. The curve represents the total SARS-CoV-2 vaccine doses (Comirnaty^®^—BNT162b2 and Spikevax^®^—mRNA-1273) required, while the bars represent the total demand for influenza vaccines (Influvac Tetra^®^, Flucelvax Tetra^®^, Vaxigrip Tetra^®^, Fluenz Tetra^®^, and Fluad^®^).

**Figure 3 vaccines-12-00158-f003:**
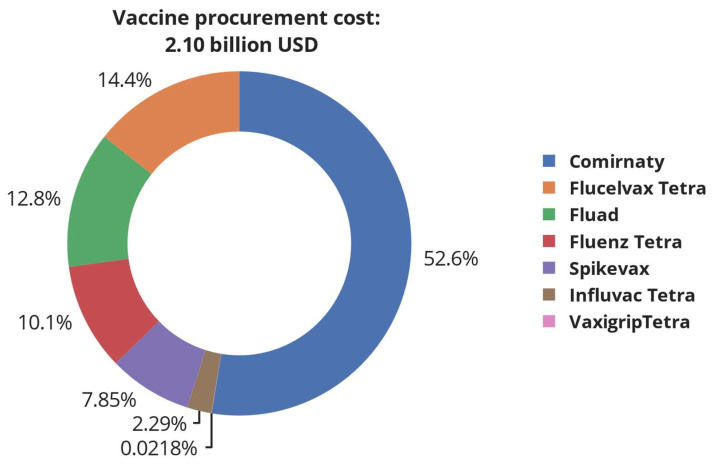
Breakdown of total cost required to procure all vaccine candidates used during the vaccination campaign in the UK. The total cost covers both SARS-CoV-2 vaccines (Comirnaty^®^—BNT162b2 and Spikevax^®^—mRNA-1273) and influenza vaccines (Influvac Tetra^®^, Flucelvax Tetra^®^, Vaxigrip Tetra^®^, Fluenz Tetra^®^, and Fluad^®^).

**Figure 4 vaccines-12-00158-f004:**
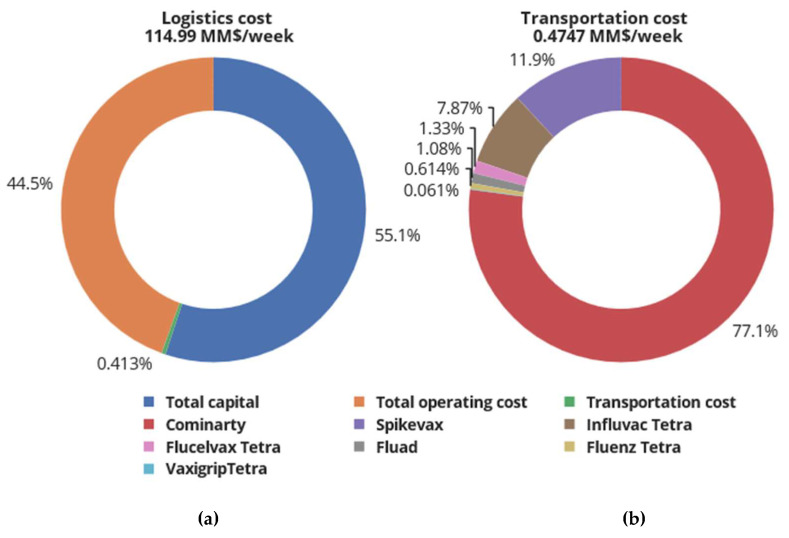
Breakdown of logistics cost and cost of transporting vaccines from manufacturing plants to vaccine administration points in the UK. Logistics cost is the sum of total capital cost, total operating cost, and total transportation cost.

**Figure 5 vaccines-12-00158-f005:**
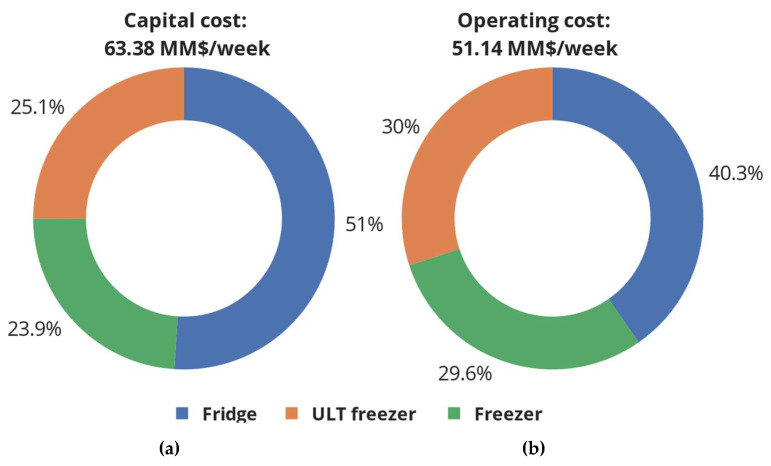
Capital and operating cost of cold chain devices installed at central warehouses and regional stores in the UK. The installed devices include fridges, freezers, and ultra-low-temperature freezers.

**Figure 6 vaccines-12-00158-f006:**
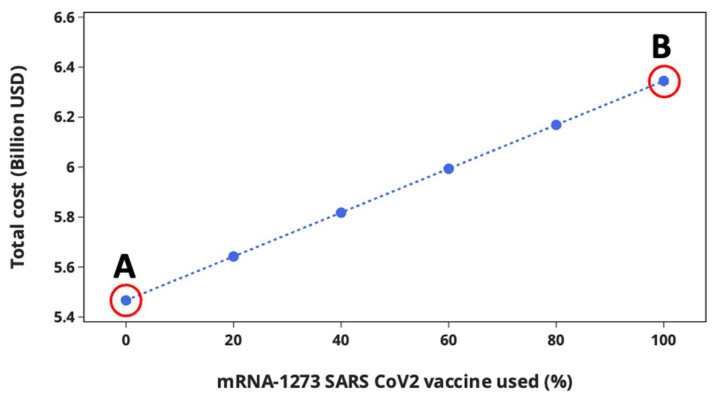
Impact of SARS-CoV-2 vaccine type on total vaccination cost. Scenario A denotes when BNT162b2 SARS-CoV-2 vaccine is used to cover 100% of the vaccination campaign, while Scenario B denotes when 100% mRNA-1273 is used.

**Figure 7 vaccines-12-00158-f007:**
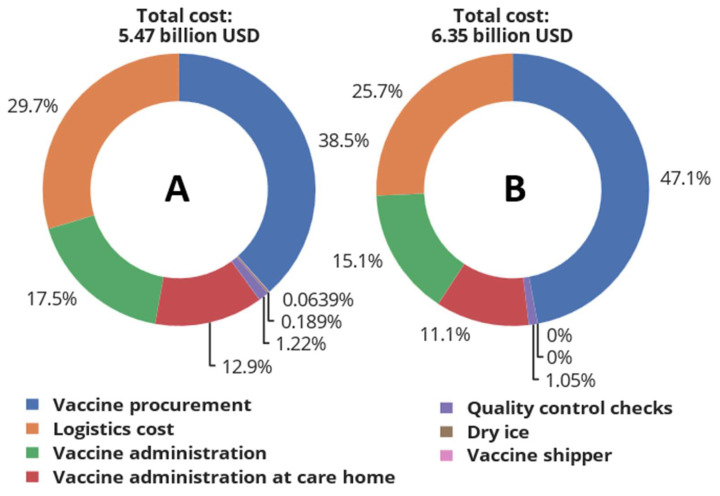
Cost breakdown for Scenarios A and B. Scenario A denotes when BNT162b2 SARS-CoV-2 vaccine is used to cover 100% of the vaccination campaign, while Scenario B denotes when 100% mRNA-1273 is used.

**Figure 8 vaccines-12-00158-f008:**
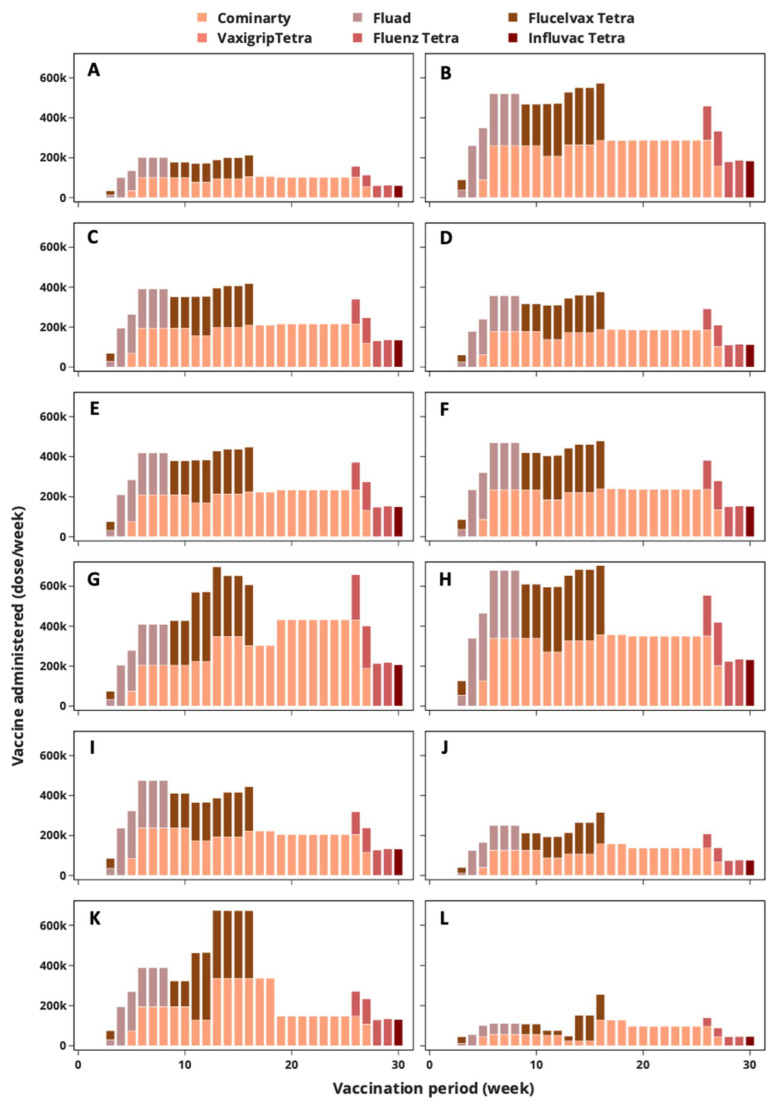
SARS-CoV-2 and influenza vaccines administered at various locations across the UK: (**A**) North East, (**B**) North West, (**C**) Yorkshire and the Humber, (**D**) East Midlands, (**E**) West Midlands, (**F**) East of England, (**G**) London, (**H**) South East, (**I**) South West, (**J**) Wales, (**K**) Scotland, (**L**) Northern Ireland.

**Table 1 vaccines-12-00158-t001:** Total cost of SARS-CoV-2 and influenza vaccination in the UK. This includes the cost of vaccine shippers, dry ice, administration of vaccines, vaccine procurement, quality control, and logistics.

Item	Value	Unit
Cost of vaccine shipper	0.0035	Billion USD
Cost of dry ice	0.0103	Billion USD
Cost of vaccinating individuals	0.9586	Billion USD
Cost of vaccinating individuals at care home	0.7027	Billion USD
Cost of vaccine procured	2.1038	Billion USD
Cost of quality control checks	0.0664	Billion USD
Logistics cost	3.4496	Billion USD
Total cost	7.2950	Billion USD

## Data Availability

Data are contained within the article and Appendix A.

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
