# Peer review of "Strategic Planning of a Joint SARS-CoV-2 and Influenza Vaccination Campaign in the UK"

_vaccines, 2024, doi:10.3390/vaccines12020158_

Round 1

Reviewer 1 Report

Comments and Suggestions for Authors

this work analyzes an important problem in an innovative way, it is well written and extremely detailed, especially in the methodological part, this could hinder proper reading by those who should be more motivated, the decision makers

in my opinion the methodological part could be reduced and the details, especially the formulas, listed in an annex

Author Response

Reviewer #1:  

This work analyzes an important problem in an innovative way, it is well written and extremely detailed, especially in the methodological part, this could hinder proper reading by those who should be more motivated, the decision makers. In my opinion the methodological part could be reduced and the details, especially the formulas, listed in an annex.

We thank the reviewer for the comments and suggestions. The formulas denoting the mathematical model of the SARS CoV2 and influenza vaccine supply chain have been moved to the supplementary material.

Reviewer 2 Report

Comments and Suggestions for Authors

General comment

The paper “ Strategic planning of joint COVID-19 booster and influenza 2 vaccination campaign in the UK is an interesting article about a novel multi-product MILP vaccine supply chain model that can be used to plan and optimise the simultaneous distribution and administration of COVID-19 boosters and influenza vaccines. The article is well written and has relevant results.

Minor comments

The authors have made an important effort to present it according to the structure of the usual scientific articles, but some aspects should be presented in a friendlier format (the entire section 2). In addition, a map with the warehouse regions should accompany it.

Specific comments

1)      Review the Abstract and add the relevant aspects of results (cost of COVID-19 vaccine versus influenza; mastery of logistics costs than the vaccine, ...).

2)      Review the methods section and present it in a more user-friendly way (pass content as supplementary material)

3)      Add a map with the warehouses

4)      Review all the text and correct small errors (example page 4 "fucntion")

Comments on the Quality of English Language

None

Author Response

Reviewer #2:  

General comment

The paper “ Strategic planning of joint COVID-19 booster and influenza 2 vaccination campaign in the UK” is an interesting article about a novel multi-product MILP vaccine supply chain model that can be used to plan and optimise the simultaneous distribution and administration of COVID-19 boosters and influenza vaccines. The article is well written and has relevant results.

We thank the reviewer for the comment.

Minor comments

The authors have made an important effort to present it according to the structure of the usual scientific articles, but some aspects should be presented in a friendlier format (the entire section 2). In addition, a map with the warehouse regions should accompany it.

We thank the reviewer for the comments and suggestions. The formulas denoting the mathematical model representing the supply chain have been moved to the supplementary material. In addition, a UK map showing the locations of manufacturing plants, warehouses, regional stores, and vaccine administration points have been presented in the supplementary material.

Specific comments

  • Review the Abstract and add the relevant aspects of results (cost of COVID-19 vaccine versus influenza; mastery of logistics costs than the vaccine, ...).

We thank the reviewer for the comments and suggestions. The abstract has been reviewed and relevant results have been incorporated accordingly.

Old abstract

“The simultaneous administration of SARS CoV2 and influenza vaccines is carried out for the first time in the UK and around the globe, in order to mitigate the health, economic, and societal impact of the respiratory tract diseases. However, a systematic approach for planning the vaccine distribution and administration aspects of the vaccination campaigns would be beneficial. This work develops a novel multi-product mixed-integer linear programming (MILP) vaccine supply chain model that can be used to plan and optimise the simultaneous distribution and administration of SARS CoV2 and influenza vaccines. Outcomes from this study reveal the budget that must made available to successfully accomplish the SARS CoV2 and influenza vaccination campaigns in addition to the various resources required ─ such as cold chain equipment (fridge, freezer, and ultra-low temperature freezer), transport devices, healthcare personnel, and doses of recommended vaccine candidates. Analysis of results shows that the choice of SARS CoV2 vaccines to be rolled-out during the vaccination campaign can have a significant impact not only on the total vaccination cost but also on vaccine wastage rate.”

New abstract

“The simultaneous administration of SARS CoV2 and influenza vaccines is carried out for the first time in the UK and around the globe, in order to mitigate the health, economic, and societal impact of the respiratory tract diseases. However, a systematic approach for planning the vaccine distribution and administration aspects of the vaccination campaigns would be beneficial. This work develops a novel multi-product mixed-integer linear programming (MILP) vaccine supply chain model that can be used to plan and optimise the simultaneous distribution and administration of SARS CoV2 and influenza vaccines. Outcomes from this study reveal that the total budget required to successfully accomplish the SARS CoV2 and influenza vaccination campaigns is equivalent to $7.29 billion, of which the procurement cost of SARS CoV2 and influenza vaccines corresponds $2.1 billion and 0.83 billion respectively. Logistics cost is equivalent to $3.45 billion and the cost of vaccinating individuals, quality control checks, and vaccine shipper and dry ice correspond to $1.66, $0.066, and $0.014 respectively. Analysis of results shows that the choice of SARS CoV2 vaccines to be rolled-out during the vaccination campaign can have a significant impact not only on the total vaccination cost but also on vaccine wastage rate.”

  • Review the methods section and present it in a more user-friendly way (pass content as supplementary material)

We thank the reviewer for the comments. The formulas denoting the mathematical model representing the supply chain have been moved to the supplementary material.

  • Add a map with the warehouses

A UK map showing the locations of manufacturing plants, warehouses, regional stores, and vaccine administration points have been added to the supplementary material. See Section S2

  • Review all the text and correct small errors (example page 4 "fucntion")

All text has been reviewed and corrected accordingly. The corrections are highlighted in red text.

Reviewer 3 Report

Comments and Suggestions for Authors

The article “Strategic planning of joint COVID-19 booster and influenza vaccination campaign in the UK” has been reviewed. SARSCoV2 has now become one more of the gang  performing during the cold season and thus increases the incidence of acute respiratory infections. Stil playing leading roles are Influenza, Sincitial Respiatory virus and Rhinovirus . The availability of preventive measures ( pharmacological and non-pharmacological) are available for SARS CoV2 and Influenza as well as immunotherapy and very soon vaccine for RSV. That gives public health tool to try and cope with the overload of consultations and hospitalizations during peak epidemic periods. Yet it is important to be able to achieve high immunization coverage especially among those at risk of complications and among those taking care of them. Simultaneous administration of seasonal SARS CoV2 and influenza vaccines will be the common issue from now on , and this paper tries to protocol the way this policy can be implemented  in the UK and around the globe.  However, a systematic approach for planning the vaccine distribution and administration aspects of the vaccination campaigns would be beneficial. This research presents a mathematical formulation that develops a novel multi-product mixed-integer linear programming (MILP) to be applied for planning and optimization.  The framework is constructed using mixed-integer linear programming (MILP) which is a mathematical description of a problem through linear inequalities and expressions.

 Outcomes from this study reveal the budget that must made available to successfully accomplish influenza and SARS CoV2 vaccination campaigns and other resources required.

The work is well written and comprehensive, although mathemical calculations and formulas fall out of my area of expertise. Therefore no comments on that. Instead, I would like to point out that

COVID-19 booster is to be best expressed as SARS CoV2 vaccine because COVID-19 is the disease not the virus and because it’s no longer a booster, the composition has been changed in order to update its composition.  It’s naming has to be in parallel with influenza vaccine ( you wouldn’t call it a booster) The composition of both vaccines will follow the same track. Each year strains of virus included in each one will have to be changed according to genetic changes of the circulating viruses.

I suggest also not using flu vaccine but influenza vaccine

Author Response

Reviewer #3:

The article “Strategic planning of joint COVID-19 booster and influenza vaccination campaign in the UK” has been reviewed.

SARS CoV2 has now become one more of the gang performing during the cold season and thus increases the incidence of acute respiratory infections. Stil playing leading roles are Influenza, Sincitial Respiratory virus and Rhinovirus. The availability of preventive measures (pharmacological and non-pharmacological) are available for SARS CoV2 and Influenza as well as immunotherapy and very soon vaccine for RSV. That gives public health tool to try and cope with the overload of consultations and hospitalizations during peak epidemic periods. Yet it is important to be able to achieve high immunization coverage especially among those at risk of complications and among those taking care of them. Simultaneous administration of seasonal SARS CoV2 and influenza vaccines will be the common issue from now on, and this paper tries to protocol the way this policy can be implemented in the UK and around the globe.  However, a systematic approach for planning the vaccine distribution and administration aspects of the vaccination campaigns would be beneficial. This research presents a mathematical formulation that develops a novel multi-product mixed-integer linear programming (MILP) to be applied for planning and optimization.  The framework is constructed using mixed-integer linear programming (MILP) which is a mathematical description of a problem through linear inequalities and expressions.

Outcomes from this study reveal the budget that must made available to successfully accomplish influenza and SARS CoV2 vaccination campaigns and other resources required. The work is well written and comprehensive, although mathematical calculations and formulas fall out of my area of expertise. Therefore no comments on that. Instead, I would like to point out that COVID-19 booster is to be best expressed as SARS CoV2 vaccine because COVID-19 is the disease not the virus and because it’s no longer a booster, the composition has been changed in order to update its composition.  It’s naming has to be in parallel with influenza vaccine (you wouldn’t call it a booster) The composition of both vaccines will follow the same track. Each year strains of virus included in each one will have to be changed according to genetic changes of the circulating viruses.

I suggest also not using flu vaccine but influenza vaccine

We thank the reviewer for the comments and suggestions. All COVID-19 booster has been replaced with SAR CoV2 vaccine. Likewise, flu vaccine has been replaced with influenza vaccine. The corrections are highlighted in red text.